# Bone Metabolic Changes and Osteoporosis During Pregnancy and Lactation: A View from Dental Medicine

**DOI:** 10.3390/ijms262110476

**Published:** 2025-10-28

**Authors:** Mai Nishiura, Haruhisa Watanabe, Atsuko Nakanishi-Kimura, Marie Hoshi-Numahata, Shinnosuke Nishimoto, Fumi Ueno, Riyu Koguchi, Ryutaro Takemoto, Yusuke Kurakane, Lang Bao, Tadahiro Iimura

**Affiliations:** 1Department of Pharmacology, Faculty and Graduate School of Dental Medicine, Hokkaido University, Sapporo 060-8586, Japan; nishimai@den.hokudai.ac.jp (M.N.); hwatanabe@den.hokudai.ac.jp (H.W.); anakanishi@den.hokudai.ac.jp (A.N.-K.); hoshi-numahata.marie@den.hokudai.ac.jp (M.H.-N.); nishi.fp@den.hokudai.ac.jp (S.N.); fueno@den.hokudai.ac.jp (F.U.); riyuk@den.hokudai.ac.jp (R.K.); den.ryutaro.takemoto@gmail.com (R.T.); yusuke.kurakane@den.hokudai.ac.jp (Y.K.); lang.bao.x8@elms.hokudai.ac.jp (L.B.); 2Department of Dentistry for Children and Disabled Persons, Faculty and Graduate School of Dental Medicine, Hokkaido University, Sapporo 060-8586, Japan; 3Department of Orthodontics, Faculty and Graduate School of Dental Medicine, Hokkaido University, Sapporo 060-8586, Japan; 4Department of Oral Medicine and Diagnostics, Faculty and Graduate School of Dental Medicine, Hokkaido University, Sapporo 060-8586, Japan

**Keywords:** PLO, bone metabolism, oral development, temporomandibular joint, orthodontic treatment

## Abstract

Pregnancy- and lactation-associated osteoporosis (PLO) is receiving increasing attention. During pregnancy and lactation, bone metabolism is dramatically changed to supply minerals to the fetus and infant, which is a major cause of PLO. Weaning of lactation is clinically a primary choice to treat lactation-induced osteoporosis since breastfeeding is a key regulator of the pathophysiology during lactation. However, breastfeeding is beneficial to the physical and mental development of infants. We also discuss the beneficial effects of breastfeeding on the oral and maxillofacial development of infants. Pharmacological treatment of PLO is also discussed. This review also discusses how dynamic regulatory changes in bone metabolism during pregnancy and lactation affect homeostasis of the temporomandibular joint (TMJ) and alveolar bone in mothers, from the perspectives of TMJ diseases and orthodontic treatment.

## 1. Introduction

Pregnancy and lactation-associated osteoporosis (PLO) is defined as osteoporosis that occurs in mothers with rapidly increasing calcium demand from late pregnancy to postpartum lactation, resulting in back pain and vertebral compression fractures in the spine [1]. The prenatal and postpartum periods are critical for the health of mothers and children. Bone fractures and related pain resulting from PLO have a significant impact on the quality of life of women. As PLO is a relatively rare disease, its pathophysiological mechanisms and effective treatment strategies have not yet been fully established [1].

The pharmacological treatment of PLO in pregnant and lactating women should be considered with caution from the viewpoint of placental passage and milk transfer. Therefore, the first choice in PLO treatment is to restrict breastfeeding to limit calcium outflow. However, premature interruption of breastfeeding can cause malnutrition and increase the risk of infection, weakness, and obesity, as indicated by the WHO [2]. In addition, weaning affects the development of the oral function in infants [3,4,5]. Thus, maternal PLO can affect the growth and development of infants, including their oral development. Therefore, PLO is an important clinical issue in pediatric dentistry. This review discusses the potential impact of weaning lactation on the oral development of infants as a treatment for PLO. The pharmacological treatment of PLO is also discussed.

Dynamic hormonal changes during pregnancy and lactation are a major cause of the pathophysiology of PLO and are likely to be associated with skeletal metabolism in the oral and maxillofacial system of the mother. Therefore, it is important to consider how changes in bone metabolism during pregnancy and lactation affect temporomandibular joint (TMJ) homeostasis and alveolar bone metabolism.

In this review, we provide an overview of the current understanding of physiological changes in bone metabolism in mothers during pregnancy and lactation. We discuss how these changes influence the TMJ and orthodontic tooth movement in mothers from the perspectives of oral surgery, medicine, and orthodontics. Hormonal changes during pregnancy and lactation can alter the oral microbiome, thus promoting gingivitis and affecting bone repair [6], but these topics are beyond the scope of this review paper.

## 2. Bone Metabolism During Pregnancy

During fetal development, approximately 80% of calcium, phosphorus, and magnesium accumulate in the developing skeleton in late pregnancy. The fetal demand for calcium and phosphorus in late pregnancy is estimated to be equivalent to 5–10% of the maternal plasma [7]. Dramatic and reversible changes in bone metabolism occur differently in late gestational mothers [1].

During late pregnancy, elevated levels of active vitamin D3 (1,25(OH)_2_D_3_) in blood circulation increase calcium absorption from the gastrointestinal tract in mothers compared with that before pregnancy [1,8] (Figure 1, Table 1). This increase in calcium absorption is driven by elevated levels of estradiol and prolactin, which upregulate maternal renal 1a-hydroxylase activity [9,10]. This is balanced by upregulated urinary calcium excretion. This suggests that the increasing calcium demand during gestation is compensated mainly by the calcium uptake in the gastrointestinal tract rather than by calcium derivation from the mother’s bone.

The secretion of parathyroid hormone (PTH) decreases during pregnancy, while the secretion of parathyroid hormone-related peptide (PTHrP) increases [7]. PTHrP is not measurable in the serum of non-pregnant women, and because of its relatively short half-life as a peptide factor, it acts as a local hormone in the developmental and homeostatic regulation of tissues, such as cartilage and mammary glands. Importantly, PTHrP levels are elevated during pregnancy because of increased production from the placenta, breasts, uterus, and embryonic tissues, and this phenomenon reduces the blood level of PTH as a hormonal regulator [7]. These findings indicate that PTHrP functions as a systemic hormone for bone and calcium metabolism, similar to PTH, which drives the unique regulation of bone metabolism during pregnancy. Therefore, upregulated PTHrP also contributes to increased calcium absorption during lactation by enhancing active vitamin D3 synthesis, similar to PTH in non-pregnant women.

## 3. Brain-Breast-Bone Axis Regulates Bone Metabolism During Lactation

During lactation, coordination of the brain-breast-bone axis of maternal calcium regulation contributes to an adequate supply of calcium required for milk production in which upregulated neural hormones such as oxytocin, prolactin and serotonin regulate bone metabolic molecules [8] (Figure 2, Table 1). The proper functioning of this brain-breast-bone axis regulatory system enables increased calcium demand during lactation. Breastfeeding is a critical driver of this axis, which is further discussed in a later section (see Section 6). In a lactating mother, the calcium uptake from the gastrointestinal tract is unchanged compared with that before pregnancy [11]. Hormonally, PTH, PTHrP, and prolactin secretion increase, while estradiol levels drop sharply [12]. The serum vitamin D level has been reported to be inversely correlated with serum PTH levels. A decrease and an increase in blood levels of 1,25(OH)_2_D_3_ and PTH, respectively, were observed after delivery [12]. These hormonal changes during lactation promote osteoclast functions. Therefore, bone resorption by osteoclasts is believed to be the main response to the rapid increase in calcium demand for milk secretion.

Increased bone resorption mainly affects the trabecular and endocortical surfaces [13,14]. In nursing women, bone turnover is thought to be promoted, since bone formation marker levels, such as procollagen 1 Intact N-terminal propeptide (P1NP) and osteocalcin, as well as bone resorption marker levels, such as collagen type 1 c-telopeptide (CTX) and n-telopeptide (NTX), have been shown to be increased [15].

In addition to bone resorption by osteoclasts on the bone surface, increased osteocyte lacunar-canalicular bone remodeling, known as osteocytic osteolysis, has been reported during lactation [16,17]. Osteocytes express osteoclast-like bone resorptive enzymes such as cathepsin K and MMP13, resulting in osteocytic bone resorption that demineralizes and resorbs bone around osteocyte lacunar and luminal networks [18,19]. These findings demonstrate that osteocytic perilacunar bone resorption and osteoclastic bone resorption participate in changes in bone metabolism during lactation. In animal model studies using rabbits and rats, the administration of teriparatide, an active peptide of human PTH, changed the size of osteocyte lacunae [20,21]. Therefore, augmented PTH and PTHrP levels during lactation are likely to regulate osteocytic perilacunar bone remodeling directly.

It has also been shown that osteocytes respond to mechanical forces and regulate the activity of bone surface cells, which affects overall bone turnover. It is important to determine how changes in lacunar volume due to osteocytic osteolysis during lactation and load changes that occur during pregnancy and postpartum affect the balance of bone metabolism through the regulation of osteoclasts and osteoblasts. These dramatic changes in bone metabolism in lactating women are likely to be associated with reduced material properties and mechanical stress of the bone, as suggested by animal models [22,23], although no direct measurement of bone strength in women has been reported. Nevertheless, nursing women do not typically have fractures, except in very rare cases of PLO.

## 4. Physiological Recovery of Bone Metabolism After Pregnancy and Lactation

Bone loss associated with pregnancy and lactation usually returns to the pre-pregnancy state within a few months [1]. The anabolic phase of bone after lactation is thought to play a major role in metabolic recovery. During the bone anabolic phase, the menstrual cycle resumes, which induces estradiol by prolactin and decreases PTHrP levels. These hormonal changes contribute to the induction of osteoclast apoptosis and decrease in bone resorption [24]. Bone formation during the anabolic phase is rapidly accelerated by the induction of osteoblast progenitor cell differentiation into osteoblasts in the bone marrow [25]. Therefore, in the post-lactation phase, bone formation overcomes bone resorption, which is due to dramatic hormonal changes. The changes that occur because of osteocytic osteolysis are also reversed. Osteocytes rapidly lose their osteoclast-like properties and return to an osteoblast-like phenotype with the ability to remineralize lacuna; thus, the hypertrophic osteocyte lacuna fully returns to its baseline volume approximately one week after weaning [19,23].

## 5. PLO

Osteoporosis is a metabolic bone disease characterized by decreased bone mass and quality, which makes bones fragile and easily broken. Osteoporosis can be classified as primary osteoporosis, such as age-related osteoporosis and postmenopausal osteoporosis, with a rapid decrease in estrogen levels, and secondary osteoporosis caused by endocrine factors, such as hyperthyroidism and Cushing’s syndrome, or drug-related factors, such as long-term steroid use, and consequences of digestive diseases, such as gastrectomy. Furthermore, PLO occurs in mothers with a rapidly increasing calcium demand from late pregnancy to postpartum lactation (Figure 3). PLO can cause back pain and vertebral compression fractures in the spine, significantly affecting the quality of life of the mother and child before and after childbirth.

The incidence rate of PLO has been reported to be 4–8 cases per million. However, a recent report by Kasahara et al. in 2024 estimated the incidence rate in Japan to be 460 cases per million. Although the number of PLO patients appears to have rapidly increased, it can be considered that potential PLO patients who have not been previously diagnosed are being increasingly frequently and precisely diagnosed with PLO because PLO has become more widely recognized [26].

Many risk factors for PLO have been proposed, such as a low bone mineral density (BMD), low body mass index (BMI), infrequent exercise, ovulatory disorders, high maternal age, and a strong family history of osteoporosis [1,12,26] (Figure 3). Although PLO is a relatively rare disease, awareness of PLO has increased in recent years, and it is important to understand the pathophysiology of PLO and establish a treatment for the disease.

A large case–control study showed that pre-pregnancy low BMD, reduced physical activity in childhood or adolescence, and dental problems in childhood increase the likelihood of PLO [27]. In recent years, particularly among young women, an increase in acceptance of the societal value that “slender is beautiful” has been proposed to contribute to the increasing occurrence of PLO. An excessive desire to be slender, like fashion models and influencers, through fashion trends and social media, is accelerating excessive dieting behavior [28]. Excessive dieting can be associated with low pre-pregnancy BMD and ovulatory disorders [29]. Ovulatory disorders, including absent, scanty, and rare menstruation, as well as anovulation, are linked to PLO occurrence, which is not unexpected, as low estrogen levels result in net bone loss [26].

A family history as a background of PLO suggests that genetic factors also play a role in its pathogenesis [30,31]. Commonly observed genetic variants in patients with PLO have been reported to involve WNT signaling-related genes (*LRP5* and *WNT1*), osteogenesis imperfecta and dentinogenesis imperfecta-related genes (*COL1A1* and *COL1A2*), and hypophosphatemia-related genes (*ALPL* and *SLC34A3*) [32]. Notably, these genes are also involved in bone metabolism and primary osteoporosis and are not specific to PLO.

## 6. Breastfeeding and Development of PLO in Lactating Mothers

The magnitude of bone loss during lactation was positively correlated with the amount of milk produced (Figure 2). Women nursing twins or triplets experience greater bone loss than women nursing one baby [33,34,35]. Exclusive breastfeeding causes greater bone loss than intermittent breastfeeding [35]. Furthermore, a longer breastfeeding duration results in greater bone loss [34,36]. A clinical study compared groups of fully breastfed and formula-fed infants [12]. In the breastfed group, bone resorption markers, such as serum TRACP5b, PTH, uNTX, and TG, were all significantly increased. In contrast, bone formation markers, such as osteocalcin, did not change markedly with breastfeeding. These observations suggest that increased bone resorption occurs only when breastfeeding. No significant difference in BMD was observed when comparing the breastfed and formula-fed groups. However, when comparing non-fracture and fracture groups observed at the lumbar spine and femoral neck, a significant decrease in BMD was observed. Therefore, decreased BMD due to breastfeeding is a critical risk factor for the development of PLO in feeding mothers. These findings support the clinical decision that early weaning from breastfeeding is often the first choice of treatment.

## 7. Breastfeeding and Infant Development, Including the Maxillofacial System

As PLO develops during pregnancy and lactation, treatment strategies including the use of anti-osteoporosis drugs must be carefully considered. As discussed above, early weaning from breastfeeding can be the first choice of PLO treatment in feeding mothers; however, this leads to concern about the impact on the infant because the lactation period is not only a period of nutritional support but also an important time for the newborn to acquire physiological functions through interaction with the mother. Furthermore, recent reports have mentioned that suckling movements are related to the development of the oral function in infants [3,5] (Figure 4).

Suckling is a dynamic process that promotes jaw-oral movements in infants, leading to the proper acquisition of coordinated movements of the lips, tongue, and jaws. Specifically, the infant first latches on to the breast and nipple and draws the nipple tip down to the hard-soft palate junction. Such feeding behavior requires coupling between periodic movements of the infant’s jaws and undulation of the tongue, known as “peristalsis”. In addition, mandibular and peristaltic movements have been shown to generate constant negative pressure in the infant’s oral cavity to induce milk ejection [37,38].

It has also been reported that long-term bottle feeding may lead to malocclusion of the primary dentition. Yonezu et al. reported that bottle-feeding may lead to incompetent lip seal and non-nutritive sucking habits (pacifier and finger sucking), thus affecting occlusal characteristics in primary dentition [39]. Chen et al. reported that a shorter duration of breastfeeding was indirectly associated with the development of permanent dentition and occlusion [3]. Children breastfed for less than six months were four times more likely to develop the habit of sucking pacifiers than those breastfed for six months. Prolonged finger sucking habits increase the probability of an anterior open bite, and pacifier habits are associated with excessive overjet and the absence of developmental space in the lower arch [3]. These findings suggest that the sucking mechanism differs greatly between artificial nipples and the breast and may contribute to the development of the maxillofacial system in infants.

Breastfeeding is also an important event in the development of the microbiome in infants, which can be associated with oral infectious diseases, including periodontitis [40]. This subject is beyond the scope of this review.

## 8. Pharmacotherapies for PLO

Drug administration to pregnant and nursing women is often contraindicated or not recommended in principle because of teratogenicity caused by placental passage and adverse events in infants caused by breast milk transfer. Therefore, the establishment of safe treatment methods is required. The first choice of treatment during the lactation period is to stop calcium efflux by breastfeeding. The drug therapy used in PLO patients may include vitamin D and calcium replacement therapy, as well as osteoporosis medications, such as teriparatide (TPTD) and bisphosphonates (BPs) [2]. The effective treatment of PLO is controversial, as there are few reported cases in clinical trials.

A previous clinical study in which mothers were treated with calcium replacement therapy reported a slight increase in lumbar spine BMD in mothers with moderate dietary calcium intake. However, this therapy did not significantly affect the overall BMD to compensate for the difference between nursing and non-nursing mothers [33].

BPs are synthetic compounds with a common phosphorus–carbon–phosphorus bond and have a high affinity for calcium hydroxyapatite in the bone. They are a class of drugs that prevent loss of bone density to treat osteoporosis. There are several concerns regarding the use of BPs in PLO patients, including the lack of long-term results with BPs in premenopausal females and placental passage of BPs accumulated in the bone [41]. In a mouse model study, administration of low-dose alendronate to lactating mother mice twice a week for two and four weeks showed increased BMD in the mothers without affecting the pups [42]. If applicable to humans, BPs may be able to protect mothers during pregnancy and lactation from bone loss and fragility without affecting the fetus or infants.

Denosumab is an anti-RANK ligand (RANKL) antibody that blocks the differentiation and function of osteoclasts, thus categorized as an anti-bone resorptive agent as well as BPs. Denosumab is an attractive drug for treating osteoporosis owing to its quick offset time. However, it has to be acknowledged that discontinuation of denosumab causes rebound and sudden stimulation of bone resorption because of accumulated osteoclast precursors in the treated bone [43,44,45]. Denosumab has a 25.4-day half-life, resulting in concentrations gradually decreasing over 4–5 months, not accumulating directly in bone like BPs, which may be an appealing point in women with childbearing potential. However, the benefits of dramatic augmentation of BMD and potential rebound-related bone fractures after drug cessation have to be well considered. A study using pregnant mice demonstrated that administration of anti-RANKL antibodies to mice in the late stage of pregnancy resulted in adverse events in their neonatal offspring [46]. This suggests that administering denosumab to pregnant women may affect fetal development and growth after birth.

Teriparatide (TPTD) is a human parathyroid hormone (1–34) with an anabolic effect on bones and is frequently the preferred choice for treating women with PLO [47]. PLO patients with vertebral fractures treated with TPTD for 24 months reportedly show increased BMD in the lumbar spine, femoral neck, and hip [27,48]. In addition to its increasing effect on BMD, clinical studies have demonstrated that chronic skeletal pain is relieved by TPTD treatment [49,50,51]. Supportively, this anti-pain effect is demonstrated to be exerted directly through peripheral sensory neurons in ovariectomized rats [52]. The pain-relieving effect of TPTD is beneficial for PLO patients with pain and may not require another pain drug. TPTD is rapidly absorbed and has a half-life of approximately 1 h; therefore, if pregnancy is desired after treatment, the effect on the fetus is expected to be minimal if quit beforehand [53]. Furthermore, a clinical study demonstrated that PLO patients treated with TPTD maintained their BMD even 12 months after cessation of treatment [54]. This study suggests that TPTD should be the preferred therapy for patients with PLO planning to have another pregnancy [55,56]. PTH and PTHrP play key roles in bone metabolism during pregnancy and lactation. It is necessary to elucidate how exogenous PTH1R agonists, such as TPTD and abaloparatide (a PTHrP derivative), affect bone metabolism in pregnant or nursing women.

## 9. Consideration of Temporomandibular Joint Disorders During Pregnancy and Lactation

Hormonal changes during pregnancy and lactation may impact the TMJ and associated tissues [57]. TMJ disorders encompass a variety of clinical conditions affecting the TMJ and its associated muscles, such as jaw discomfort, dysfunction, earache, facial pain, and headaches. These factors mainly affect women of reproductive age [58,59]. In severe cases of TMJ disorders, such as degenerative joint disease of the temporomandibular joints (DJD-TMJ), which involves progressive condylar resorption (PCR) and idiopathic condylar resorption (ICR), a clinical study reported that DJD-TMJ mainly affects women of both reproductive and perimenopausal ages [60].

Fluctuating levels of estrogen during pregnancy and post-gestation have been discussed as key regulators of oral diseases, including TMJ disorders, since estrogen is essential for the structural maintenance of joints and bones [6]. A clinical study reported that postmenopausal women with osteoporosis had a higher risk of developing TMJ disorders [61]. A study using rabbits demonstrated that estrogen deficiency did not cause mandibular resorption but did carry a risk of anterior displacement of the joint [62]. These clinical and animal studies indicate that an appropriate level of estrogen is essential for TMJ homeostasis.

However, a systematic review and meta-analysis found no significant differences in the prevalence of TMJ disorders between pregnant and non-pregnant women of reproductive age [63]. A recent case–control cross-sectional study suggested that pregnancy is neither a risk nor protective factor for temporomandibular disorder (TMD) [64]. How hormonal and bone metabolic changes other than estrogen during pregnancy and lactation affect TMJ disorders remains unclear.

Studies have shown that TMJ-related pain decreases while TMJ relaxation increases during pregnancy [59]. An experimental study has demonstrated that high levels of estrogen and progesterone have antinociceptive properties [65]. A prospective study showed that TMJ-related pain and other symptoms appear to improve over the course of pregnancy [66]. These findings suggest that elevated estrogen levels during pregnancy reduce orofacial pain and that reduced estrogen levels during lactation may promote chronic pain [57].

Accumulated clinical and preclinical animal model studies have demonstrated that pharmacological administration of PTH not only protects bone and joint degeneration but also reduces chronic skeletal pain [52,67,68,69]. It is likely that dynamic changes in PTH and PTHrP levels during lactation and pregnancy are involved in the pathophysiological control of the TMJ and orofacial pain. It will be relevant to conduct clinical and preclinical studies to determine how PTH and PTHrP levels are associated with TMJ disorders and orofacial pain.

## 10. Consideration of Orthodontic Tooth Movement During Pregnancy and Lactation

Dynamic changes in the hormonal regulation of bone metabolism during pregnancy and lactation are likely to affect the homeostatic regulation of maxillofacial bones. Because of the increasing demand for orthodontic treatment for aesthetic reasons, it is important to consider how bone remodeling induced by orthodontic tooth movement (OTM) is affected in pregnant and feeding mothers [57]. During pregnancy, upregulated serum levels of estrogen and PTHrP are key regulators of bone remodeling (Figure 1, Table 1), which lays the foundation for the rate of OTM. It positively regulates bone formation by promoting osteogenic precursors and osteoblast differentiation while suppressing osteoclast formation and function. Therefore, estrogen is fundamentally favorable for bone anabolism, which is supported by the fact that estrogen deficiency causes osteoporosis [69]. PTHrP upregulated to detectable levels in serum is a characteristic feature of bone metabolism during pregnancy and lactation and is likely to function like PTH through its common receptor of PTH1R in osteoblast lineage cells. Since PTH stimulates both bone formation and resorption in trabecular and cortical bones [21,70,71,72], augmented PTHrP during pregnancy is suggested to promote bone turnover, although downregulated PTH may compensate for the over-function of PTHrP.

Seeing the upregulated serum levels of estrogen and PTHrP during pregnancy, the acceleration of OTM sounds reasonable. Animal model studies tested the rate of OTM during pregnancy [73,74,75], which found no statistically significant difference in the rate of OTM between the pregnant and non-pregnant groups. In addition to estrogen and PTHrP, many other hormonal and biological factors are involved in bone remodeling during pregnancy, which may alter the rate of OTM. Verification with large-scale clinical studies will be needed.

Bone metabolism during lactation is characterized by decreased estrogen and increased PTHrP and PTH levels (Figure 2, Table 1), which enhances bone turnover [76]. A mouse study observed that lactation resulted in a significantly increased rate of OTM compared to that in the non-lactating group [77]. However, large-scale clinical studies and animal studies are needed to verify this observation [75,78].

## 11. Conclusions and Perspective

During late pregnancy and lactation, dramatic changes occur in maternal bone metabolism, resulting in an adequate supply of calcium required for fetal development and milk production [8]. Although PLO is relatively rare, its prevention and treatment are critical for proper infant development and improving the maternal quality of life. Although accumulated evidence suggests early weaning from breastfeeding as a primary choice to treat PLO in lactating mothers, this often burdens mothers’ mental health and may affect infant development. Therefore, the establishment of a safe pharmacotherapy is required. It is also worth noting that dramatic changes in maternal bone metabolism during pregnancy and lactation may affect the homeostasis of the TMJ and alveolar bone in mothers. Therefore, it may be an effective approach to confirm the mother’s medical history of PLO, whether the child is breastfed or bottle-fed, and the age of weaning of the child as a medical interview to monitor oral and maxillofacial health both in children and mothers. The strength of this review is proposing the relevance of knowing the changes in maternal bone metabolism for dentists as well as related medical fields. However, there are still some limitations in this subject. Large-scale clinical studies and accumulating further mechanical analysis using animal studies are needed. We still need a life-span view of premenopausal and postmenopausal pathophysiology of bone metabolism [79].

## Figures and Tables

**Figure 1 ijms-26-10476-f001:**
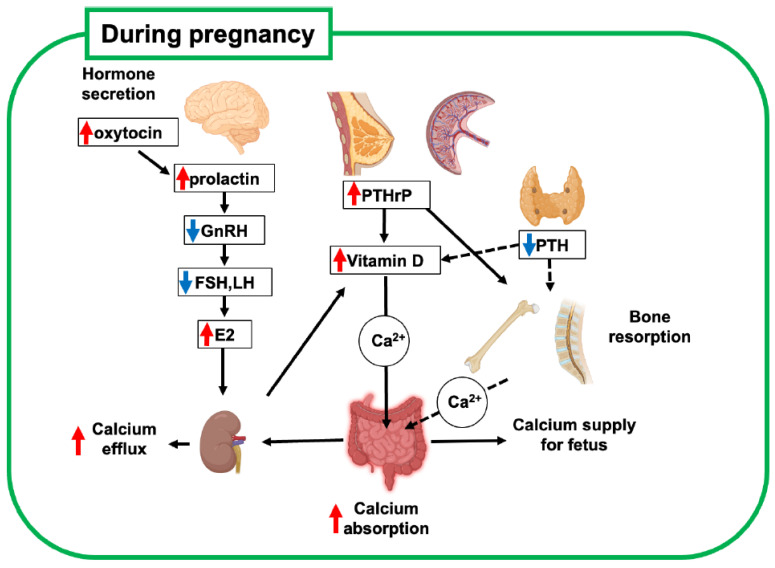
Schematic diagram of bone metabolic changes in pregnant mother. Increased estrogen and participation of PTHrP in systemic regulation are key regulators of bone metabolism and mineral supply to the fetus. Red and blue arrows indicate upregulated and downregulated hormones, respectively. This figure was created with BioRender.com (accessed on 25 September 2025). PTH, parathyroid hormone; PTHrP, parathyroid hormone-related peptide; GnRH, gonadotropin-releasing hormone; FSH, follicle-stimulating hormone; LH, luteinizing hormone; E2, estradiol.

**Figure 2 ijms-26-10476-f002:**
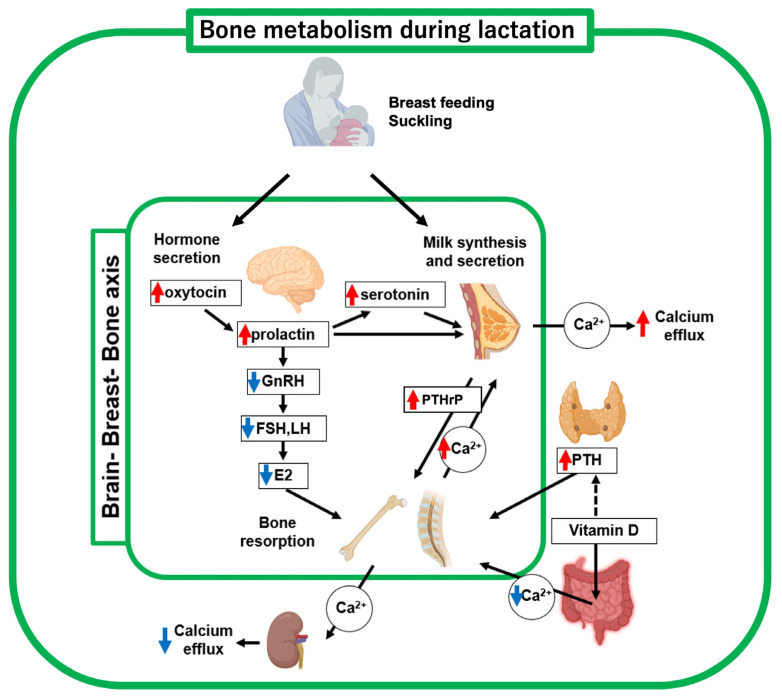
Schematic diagram of bone metabolic changes in lactating mother. Decreased estrogen and increased PTHrP and PTH levels, respectively, are key regulators of bone metabolism and mineral supply to breast milk. Suckling (breast-feeding) upregulates oxytocin, prolactin, and serotonin, thus driving the brain-breast-bone axis. Red and blue arrows indicate upregulated and downregulated hormones, respectively. This figure was created with BioRender.com (accessed on 25 September 2025).

**Figure 3 ijms-26-10476-f003:**
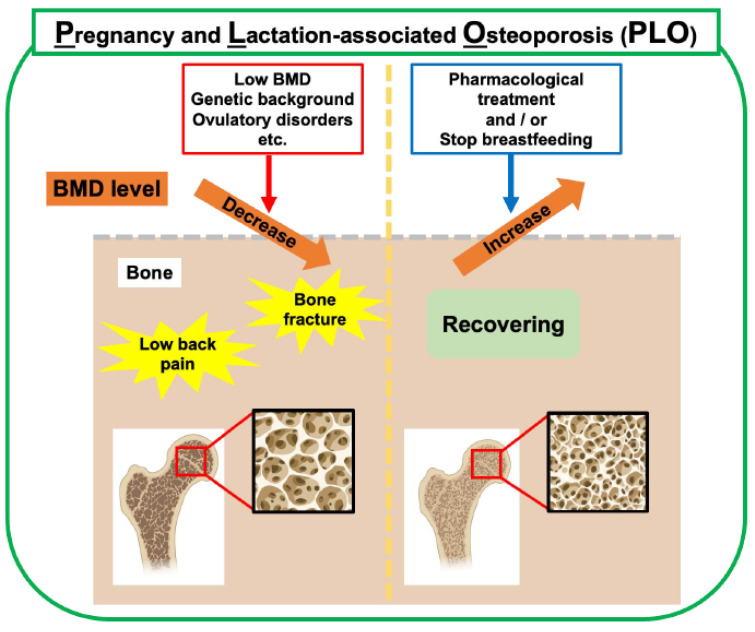
Risk factors and management of PLO. In addition to bone metabolic changes during pregnancy and lactation, low BMD, genetic background, and ovulatory disorders are risk factors. Weaning breastfeeding is the primary choice for treating PLO in lactating mothers. Pharmacological treatment with anti-osteoporosis drugs is clinically used to treat PLO. This figure was created with BioRender.com (accessed on 25 September 2025).

**Figure 4 ijms-26-10476-f004:**
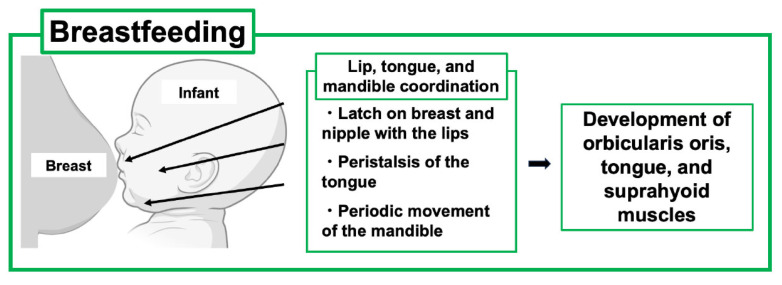
Effects of breastfeeding on the development of the oral maxillofacial system of the child. Breastfeeding is beneficial for infant development of oral and maxillofacial systems through lip, tongue, and mandible coordination. This figure was created with BioRender.com (accessed on 25 September 2025).

**Table 1 ijms-26-10476-t001:** Physiological changes in the mother during pregnancy and lactation. PTH: Parathyroid hormone, PTHrP: Parathyroid hormone-related peptide, GnRH: Gonadotropic releasing hormone, FSH: Follicle-stimulating hormone, LH: Luteinizing hormone, E2: estradiol, CTX: collagen type 1 c-telopeptide, NTX: collagen type 1 n-telopeptide, P1NP: procollagen type 1 amino-terminal propeptide, BMD: bone mineral density.

	Pregnancy	Lactation
Brain	↑ oxytocin, prolactin	↑ oxytocin, prolactin
	↓ GnRH, FSH, LH	↓ GnRH, FSH, LH
Parathyroid gland	↓ PTH	↑ PTH
Breast	↑ PTHrP	↑ PTHrP
Intestine	↑ calcium absorption	Normal calcium absorption
Kidney	↑ urinary calcium excretion	↓ urinary calcium excretion
Ovary	↑ E2	↓ E2
Blood serum	↑ CTX, NTX	↑ CTX, NTX
	↑ P1NP	↑ P1NP, osteocalcin
Bone	↓ BMD spine up to −4.6%	↓ BMD spine up to −7%
	↓ BMD hip up to −4.2%	↓ BMD hip up to −3%

## Data Availability

No new data were created or analyzed in this study. Data sharing is not applicable to this article.

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
