# Peer review of "Bone Metabolic Changes and Osteoporosis During Pregnancy and Lactation: A View from Dental Medicine"

_ijms, 2025, doi:10.3390/ijms262110476_

Round 1

Reviewer 1 Report

Comments and Suggestions for Authors

Comments on the Quality of English Language

None

Author Response

Response 1: Thank you for pointing this out. We agree with this comment. Therefore, we have fully edited sections 8. Pharmacotherapies for PLO (page 9-), and 10. Consideration of Orthodontic Tooth Movement During Pregnancy and Lactation (page 12-) to avoid redundant description and make them more streamlined.

Comments 2: Figures and tables: Ensure all figures (BioRender-based) are original and comply with copyright rules. Tables should be better aligned with clinical implications.

Response 2: Agree. We have, accordingly, revised the figure legends. We have not changed the table 1, since we think it is clear and simple enough to bring the message.

Comments 3: Language and style: Minor grammatical and stylistic errors require editing for smoother readability.

Response 3: Whole text was checked and edited by a professional scientific editor. 

Comments 4: While the review is valuable, the TMJ and orthodontic sections should be condensed, focusing on clinical evidence rather than repeating animal data extensively.

Response 4: Thank you for this comment. As we mentioned in the Response 1, we have fully edited sections 8 and 10.

Comments 5: Needs to be more precise, with clear take-home messages for clinicians and researchers.

Response 5: Thank you for pointing this out. We have, accordingly, revised the 11. Conclusion and Perspective (in page 11).

Reviewer 2 Report

Comments and Suggestions for Authors

In this review, the authors provide an overview of the current understanding of physiological changes in bone metabolism in mothers during pregnancy and lactation. They discuss how  these changes influence the TMJ and orthodontic tooth movement in mothers from the  perspectives of oral surgery, medicine, and orthodontics.

The authors conduct a narrative review of the role of pregnancy and breastfeeding in the onset of osteoporosis. The review is comprehensive and interesting, with particularly useful infographics that help to understand the problem.

It would be interesting if the authors included:

- The methodology used to carry out the review

- Its limitations and strengths

- A section reviewing the consequences of osteoporosis during pregnancy and breastfeeding on postmenopausal osteoporosis

Author Response

Comments 1: In this review, the authors provide an overview of the current understanding of physiological changes in bone metabolism in mothers during pregnancy and lactation. They discuss how these changes influence the TMJ and orthodontic tooth movement in mothers from the perspectives of oral surgery, medicine, and orthodontics.

The authors conduct a narrative review of the role of pregnancy and breastfeeding in the onset of osteoporosis. The review is comprehensive and interesting, with particularly useful infographics that help to understand the problem.

It would be interesting if the authors included:

- The methodology used to carry out the review

- Its limitations and strengths

- A section reviewing the consequences of osteoporosis during pregnancy and breastfeeding on postmenopausal osteoporosis

Response 1: Thank you very much for pointing these out. We totally agree with these. We have not described the methodology of this review, since this is not a systemic review of well specified subject. What we are aiming is rather to point out the relevance of an inter disciplinary understanding between dentists and related medical fields. Limitations and strengths are accordingly added to the section of 11. Conclusion and Perspective (page 12). Thanks for this comment. As pointed out by the reviewer 2, it is important to understand the association between premenopausal and postmenopausal bone pathophysiology. However, few findings on this point have been reported. Therefore, we have mentioned this also in the section of 11. Conclusion and Perspective.

Round 2

Reviewer 1 Report

Comments and Suggestions for Authors

Well done 

Reviewer 2 Report

Comments and Suggestions for Authors

The questions have answered by the authors